# BALANCED LEARNING WITH TOKEN SELECTION FOR FEW-SHOT CLASSIFICATION

## ABSTRACT

In recent years, patch-based approaches have shown promise in few-shot learning, with further improvements observed through the use of self-supervised learning. However, we observe that the mainstream object-oriented approach focuses mainly on the salient part of the subject and also ignores the non-annotated part of the image. Based on the assumption that any patch of the image is beneficial to learning, we present an end-to-end learning framework, which reconsiders the whole image from a multi-level perspective. The learning of annotated subjects involves Direct Patch Learning (DPL) to promote balanced learning of different features, and Gaussian Mixup (GMIX) to provide extra mixed patch-level labels. As for the non-annotated part, we utilize a cascading token selection strategy along with self-supervised learning to better utilize knowledge in the background in the current context by learning the consistent representation of different views from the same image. Finally, in inductive few-shot learning, our method outperforms many previous methods and achieves new state-of-the-art performance. Furthermore, it provides an insight that non-annotated parts are also favorable for few-shot learning. As an ablation study, the effectiveness of each designed component is verified and the mechanism of how our method outperforms the baseline is shown both quantitatively and visually.

## 1 INTRODUCTION

Few-shot Learning (FSL) is a highly challenging task, which aims to adapt to new tasks using a very small amount of labeled data. In recent years, many methods (Finn et al., 2017; Vinyals et al., 2016a; Tian et al., 2020a; Chen et al., 2019; Jamal & Qi, 2019; Hao et al., 2019; Li et al., 2019; Qiao et al., 2019; Sun et al., 2019; Rodríguez et al., 2020; Jelley et al., 2022) have been proposed to tackle this problem. Most few-shot methods contain two stages, meta-training, and meta-testing. After pre-training a backbone on the base set in meta-training, the method's performance on novel classes is evaluated on lots of few-shot tasks during meta-testing. Few-shot learning can be categorized as transductive and inductive methods. Their difference is that transductive methods add the novel set as unlabeled data to the base set. This paper will focus on the more general inductive method.

Some recent studies (Hiller et al., 2022; Zhang et al., 2020; He et al., 2022; Lifchitz et al., 2019; Huang et al., 2021) have shown that patch-based methods benefit few-shot learning. DeepEMD (Zhang et al., 2020) regarded each patch as a component of an object. The similarity used for classification is calculated by optimal matching between patches. However, the Hungarian algorithm for solving the optimal matching is computationally expensive. Densecls (Lifchitz et al., 2019) trained each local patch using an image-level label to promote consistent predictions across different patches. Overall, it has been argued in previous object-oriented few-shot work that a complete classification task can be achieved using only the part in which the object resides. However, certain patches, so-called background patches, might contain overlapping objects and richer semantic information, which the model did not entirely leverage during training, further limiting its performance.

Based on the thoughts above, Tokmakov et al. (2019) introduced background attributes in a limited way to enable background learning. However, this research trend has not resulted in a viable end-to-end learning model.

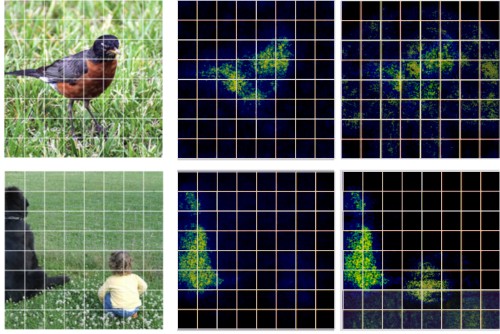

Input     Without Cascading    With Cascading

Figure 1: Guided-backpropagation (Springenberg et al., 2015) visualization of our strategy. The second column confirms the phenomena mentioned by motivation in Sec.3.1 that there is indeed ignorance of annotated parts within objects (body and tails) and non-annotated parts beyond objects (grassland). These phenomena were mitigated by our proposed strategy.

To tackle these concerns, we propose a multi-stage end-to-end framework, named Cascading Patch-Wise Network. The overall framework comprises $n$ successive token selections along with self-supervised learning loss. The token selection strategy separates tokens into "top tokens" and "bottom tokens", based on the ranking scores. In foreground learning (basically the top tokens after the first selection), we propose two methods, Direct Patch Learning (DPL) and Gaussian Mixup (GMIX). DPL operates on particular local foreground patch tokens, while GMIX offers patch-level labels to DPL by mixing different patches. In contrast, we utilize self-supervised learning to each level of bottom tokens to further learn the image's structural information and increase data utilization efficiency, thereby enhancing the model's robust representation capabilities.

In summary, our main contributions are:

- An end-to-end Cascading Patch-Wise Network for fully utilizing the contextual information for few-shot learning is proposed. Based on such a network, our method makes significant improvements on its baseline and proves the potential value of non-annotated parts.

- The token selection strategy is implemented to divide the learning process into "top tokens" and "bottom tokens". The employment of self-supervised learning to acquire knowledge from the "bottom tokens" brings effective utilization of the available data.

- Direct Patch Learning along with Gaussian Mixup is utilized to balance the learning of diverse features and improve the representational capacity of local tokens. By demonstrating the visualization results of local tokens' activation areas, we validate the efficacy of Direct Patch Learning.

## 2 RELATED WORK

**Meta-learning.** Meta-learning is the dominant paradigm in few-shot learning, which makes the model generalize to new tasks better by constructing multiple learning tasks. Researchers have proposed many meta-learning methods (Finn et al., 2017; Koch et al., 2015; Snell et al., 2017a; Vinyals et al., 2016a; Oreshkin et al., 2018b). Some other methods (Chen et al., 2019; Lee et al., 2019b; Tian et al., 2020b; Mangla et al., 2020) focus on pre-training a better backbone, and the few-shot classification problem was solved through a linear classifier or metric learning. In this paper, we pay more attention to training an efficient backbone and improving the generalization ability of the model to novel categories.

**Patch-based Method.** LMPNet (Huang et al., 2021) considered a local token as a local descriptor, calculates the distance between each pair of tokens in two images, and takes the average distance of all pairs as the matching score for the images. DeepEMD (Zhang et al., 2020) followed a similar idea as LMPNet (Huang et al., 2021), but it computes the matching score for image tokens using optimal matching. It needs to solve an optimal matching problem during training, which is computationally expensive. Densecls (Lifchitz et al., 2019) trained each local patch with an image-level label to encourage the same prediction over different patches. Our work here takes a different approach, aiming to strengthen the representational ability of tokens and better exploit knowledge from given limited data.

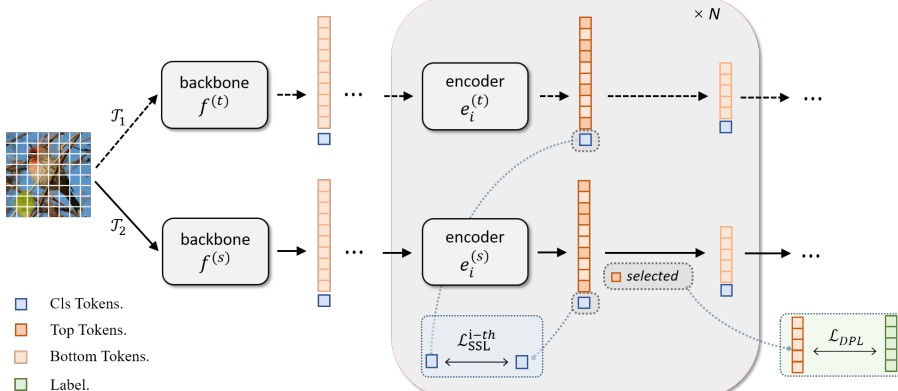

Figure 2: Overall pipeline. The illustration of the multi-stage learning framework that utilizes a token selection strategy to remove the top tokens in the current context. For the first encoder, the removed top tokens (foreground tokens) are utilized to perform the direct patch learning, for the following encoders, the removed tokens were discarded. For all encoders, the remaining bottom tokens were inputted to the next encoder for self-supervised learning.

**Self-supervised learning in FSL.** In papers of Gidaris et al. (2019), He et al. (2022), Hiller et al. (2022); Su et al. (2020), self-supervised methods were introduced for few-shot learning, and they indicated that supervised learning provides inferior performance compared with self-supervised methods in few-shot scenarios. We adopt the self-supervised method DINO (Caron et al., 2021), iBot (Zhou et al., 2022), DINOv2 (Oquab et al., 2023) to squeeze residual information beyond the annotated object. Finally, the three methods are compared visually and quantitatively.

**Cascading learning.** HCT (He et al., 2022) used spectral clustering to perform pooling on patches so that the model learns at different semantic levels. Similar to ours, HCT uses the framework of self-supervised learning methods to achieve SOTA performance. But our motivation and method are still different from HCT (detailed in Appendix.A).

## 3 METHOD

### 3.1 DEFINITION AND MOTIVATION

**Definition**: Few-shot classification involves dividing the dataset into two distinct sets, denoted as $\mathcal{D}_{base}$ and $\mathcal{D}_{novel}$. The former set, $\mathcal{D}_{base}$, is utilized to conduct meta-training, which is a process of training a model to learn how to learn. When testing, few-shot testing task is constructed on $\mathcal{D}_{novel}$, it contains a tuple $(\mathcal{D}_{support}, \mathcal{D}_{query})$. If $\mathcal{D}_{support}$ contains $K$ classes and each class has $N$ samples, it is called a $K$-way-$N$-shot task.

**Motivation**: 1) "Shortcut" effect of the annotated objects. For a targeted object, networks prefer discriminating it by only part of patches. In layman's terms, not only can an elephant's trunk be used as a foreground feature to judge an elephant, but so can its legs and ears. So if a balanced learning scenario is not constructed for the foreground patches, the robustness is reduced. 2) "Unused residual information" beyond the annotated objects. These parts of the image with no class labels also hold a prior probability which brings limited learning value.

**Optimisation goals**: Overall, we constructed different cascading weight redistribution strategies for the patch ranked by token selection. For the object itself, we alleviate the shortcut effect that occurs when a certain patch is too easy to be recognized. For low-ranked patches that contain even less information, a cascading iterable method is used to squeeze out the residual information.

Our overall pipeline is illustrated in Fig.2. Multiple token selection module (Sec.3.2) is used to distinguish "top tokens" from "bottom tokens". Generally, only the top and bottom tokens after the first selection can be interpreted as foreground and background. To alleviate "Shortcut" effect of the annotated object, the Direct Patch Learning (Sec.3.3) is proposed for the foreground, which is only used after the first selection. Finally, the cascading self-supervised learning strategy (Sec.3.4) will squeeze limited information patch-wisely out of the rest of the bottom tokens.

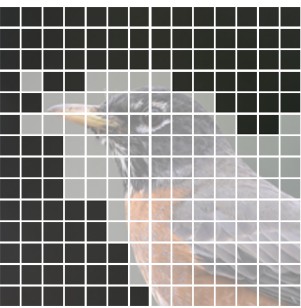

Figure 3: Patches of the first token selection. Patches with brighter masks are selected as top patches for Direct Patch Learning, while patches with darker masks are considered bottom patches that serve as input for the second selection, which is utilized for mining background information.

## 3.2 TOKEN SELECTION

Since the motivation in Sec.3.1 mentions the need to build a supervised representation of the un-labelled part of the image, it is natural to think of the currently popular self-supervised learning method, which will be also discussed in Sec.3.4 and Appendix.A. Then, further research on DINO (Caron et al., 2021), iBot (Zhou et al., 2022) and DINOv2 (Oquab et al., 2023) points out that the self-attention map of $[cls]$ tokens highlights the areas where the foreground object is located, and suppresses the background area. Inspired by this, we present a feature selection mechanism based on Random Walk that distinguishes "top tokens" from "bottom tokens" in the current context. Furthermore, we do not merely discard the bottom tokens but employ self-supervised learning to mine the structural information inherent in the image, which we find beneficial for few-shot learning.

The token selection strategy is where tokens with high relativity with $[cls]$ token are selected as top tokens for patch-level supervision training, while those with lower relativity are used as input for the next encoder to learn background representation. Fig. 3 illustrates a visualized example of the token selection process.

To select tokens that are most closely associated with the $[cls]$ token (i.e., top tokens), it is necessary to determine the degree of relevance between the $[cls]$ token and patch tokens. Inspired by the PageRank (Page et al., 1998) algorithm, we consider different tokens as states and attention scores as the probability transition matrix between states, then construct a Markov chain. Starting from the state corresponding to $[cls]$, after $s$ steps, we obtain the distribution of states $\pi_s$, formalized as follows:

$$\pi_s = \pi_{s-1}A = \pi_0 A^s, \tag{1}$$

where $\pi_0 = \{0, 0, ..., 0, 1\}$, meaning it starts from the state corresponding to the $[cls]$ token, $A$ is a probability transition matrix from the average attention score of all heads in a multi-head trans-former. In our final setting, we set $s = 3$.

Finally, we rank the probability distribution of states, removing the top half of tokens with the highest probabilities. The remaining tokens proceed to the next encoder for further learning. An intuitive explanation for this approach is that higher probabilities indicate a stronger association between the corresponding patch tokens and the $[cls]$ token. Since the $[cls]$ token encodes the features of a specific object in the current context, we exclude these tokens, allowing the next encoder to focus on learning from the remaining tokens.

Then to extract the limited information from the remaining bottom tokens, self-supervised learning methods and cascade learning strategy are applied. The process of self-supervised learning is introduced and analyzed in Sec.3.4 and Sec.4.4.

## 3.3 DIRECT PATCH LEARNING

The "Shortcut effect" is still illustrated by the example of an elephant and its trunk below. Empirically speaking, an elephant can be identified easily by its distinctive trunk alone. An object $x$ can be described as a set of features $[t_1, t_2, \ldots, t_n]$, where $t_i$ can be parameterized as a function of $x$, $[t_1, \ldots t_n] = f_\theta(x)$. Typically, in classical deep learning classification, the features are not explicitly separated, and the conditional distribution $p(y \mid x)$ is the model's direct output.

In few-shot learning, learning only the salient features of the base class during meta-training may not be sufficient as saliency implies its strong correlation with a specific base class and will not provide effective discriminative features when dealing with novel classes. Let $t_1$ represent the feature of the trunk, and $t_2$ represent the feature of the skin texture. Due to the strong correlation between the $t_1$ and the elephant, we can observe

$$1 \approx p(y = elephant \mid t_1, t_2) \approx p(y \mid t_1) \\ = p(y \mid t_1) + 0 \cdot p(y \mid t_2). \tag{2}$$

If $t_2$ is the output of the neural network, learning of $t_2$ can be weak or non-existent, resulting in overfitting on the base classes. Essentially, this scenario can be seen as weighted learning of features, which can select features according to the class feature bias, suppress weakly correlated outputs, and benefit the classification of base classes. However, as training progresses, performance on base classes tends to improve at the expense of the ability to discriminate among novel classes (Chen et al., 2021). Although data augmentation techniques such as RandCrop can reduce this risk to some extent, we aim to further reduce this risk at the model level.

In order to address this issue, we transform the original learning objective, which is $p(y|x) = p(y|t_1, t_2, ..., t_n)$, into multiple balanced weak classifiers $p(y|t_i)$. For each weak classifier, only a single feature is utilized to construct the classifier, thus the modeling objective becomes:

$$p(y \mid x) = \frac{1}{n} \sum_{i \leq n} p(y \mid t_i). \tag{3}$$

We pursue this approach for two main purposes. Firstly, it helps reduce overfitting to the base class by imposing additional constraints. Secondly, it promotes balanced learning of different features and minimizes the occurrence of "shortcuts".

The desired situation is that $t_i$ in the features set has diverse semantic information and can be separated for optimizing using Eq.3. However, for common classification tasks, features $t_i$ are highly entangled. Separately considering these features is intractable. We use the local patches' representations as $t_i$ since patches in one image naturally represent some local attributions. Specifically, we can let $t_i$ be a feature vector from a convolutional network or a token from a transformer that describes the corresponding local area.

LMPNet (Huang et al., 2021) also considered a local token as a local descriptor, however, their purpose is more similar to that of DeepEMD (Zhang et al., 2020), which used a local token to compute matching scores between two images. Our aim is to construct a feature-diverse classification model that weakens the suppression of weakly correlated features and enhances the generalization ability to novel classes. Based on Jensen's inequality and Eq.3, we define the Direct Patch Learning loss function as:

$$L_{DPL} := -\frac{1}{n} \sum_i \mathbb{E}(\log(p(y \mid t_i))) \geq -\mathbb{E}(\log(\frac{1}{n} \sum_i p(y \mid t_i))). \tag{4}$$

That is, we optimize the upper bound of cross-entropy for the unweighted feature model, making the computation tractable.

### 3.4 PATCH-WISE LEARNING STRATEGY

**Patch-level Label.** In the few-shot classification task, only the image-level annotations are available. Besides, a local patch might contain multiple overlapping objects, such as a sheep grazing on a grassland. If this patch's label is simply assigned as "sheep", the learning of grassland features would be ignored. To alleviate this problem, we propose Gaussian MixUp (GMIX). Instead of using a scalar to mix two images, we use a mixing matrix generated from Gaussian Distribution, as shown in Fig.4. Therefore there exist patches with complex mixed semantics and hard labels are replaced by soft labels.

**n-th cascading patch-wise learning strategy**. In the first stage, after token selection for the outputs from the first encoder, we obtain foreground-relevant and background-relevant patch sets $\boldsymbol{P_f}$ and $\boldsymbol{P_b}$. Patch-level supervision loss involves only the foreground tokens from the student network, denoted as $\boldsymbol{P_f^{(s)}} = (t_1, ..., t_n)$:

$$\mathcal{L}_{DPL} = -\frac{1}{n} \sum_i^n \mathbb{E}(log(p(y \mid t_i))). \tag{5}$$

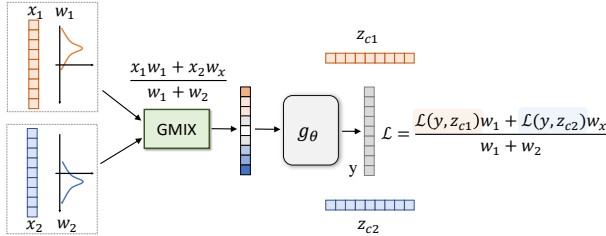

Figure 4: Gaussian MixUp for Patch-level Label. The loss function is denoted by $\mathcal{L}$, where $z$ represents the class label and $y$ denotes the network output. The GMIX algorithm generates pixel weights through two randomly generated Gaussian functions, which are utilized to blend two images and produce soft labels for Direct Patch Learning.

For $[cls]$ tokens, We used a self-supervised method and loss ($\mathcal{L}_{ssl}$) in the multiple papers like DINO (Caron et al., 2021), iBot (Zhou et al., 2022) and DINOv2 (Oquab et al., 2023) for self-supervised training. The loss used in the first stage is defined as

$$\mathcal{L}^{1^{th}} = \mathcal{L}_{ssl}^{1^{th}} + \mathcal{L}_{DPL}. \tag{6}$$

Instead of being discarded, those semantically irrelevant bottom tokens $P_b$ will be used as input for the second encoder for further mining of the rich contextual information inherent in the image. The $[cls]$ token from the $n^{th}$ stage will only be used for self-supervised loss computation (detailed in Appendix.A) as in the previous stage. Then the total loss is

$$\mathcal{L}^{n^{th}} = \mathcal{L}_{ssl}^{n-1^{th}} + \mathcal{L}_{DPL} + \alpha \mathcal{L}_{ssl}^{n^{th}}, \tag{7}$$

while $\alpha$ is the degradation hyper-parameter. We presented an overall pipeline in Algorithm 1, where we omitted GMIX for the sake of simplicity.

During testing, only the output of the first encoder is utilized, and the average of foreground tokens and $[cls]$ token is concated as the final feature vector. A cosine classifier is constructed to perform the few-shot classification task.

---

**Algorithm 1** Training pipeline

$\theta_s$, $\theta_t$ are the teacher network and student network' parameters.

1: **for** each $epoch \in [0, total\_epochs)$ **do**
2:    **for** each iteration **do**
3:       $L \leftarrow L_{DPL}$
4:       **if** $epoch > stage1\_epoch$ **then**
5:          **for** each $n^{th} \in total\_layers$ **do**
6:             $L \leftarrow L + \alpha L_{ssl}^{n^{th}}$
7:          **end for**
8:       **end if**
9:       $\theta_s \leftarrow \theta_s - lr * \frac{\partial L}{\partial \theta_s}$
10:      $z \leftarrow z - lr * \frac{\partial L}{\partial z}$
11:      $\theta_t \leftarrow \theta_t * momentum + \theta_s * (1 - momentum)$
12:    **end for**
13: **end for**

---

## 4 EXPERIMENT

### 4.1 DATASET

MiniImageNet (Vinyals et al., 2016b) is a subset of the ImageNet (Deng et al., 2009) dataset. It comprises 100 classes, out of which 64, 16, and 24 are used for training, validation, and testing, respectively. Each class in MiniImageNet consists of 600 images, resulting in a total of 600,000 images.

TieredImageNet (Ren et al., 2018) is also a subset of ImageNet and represents an extension of MiniImageNet, encompassing 600 classes, of which 351, 97, and 160 classes are allocated to the training, validation, and testing splits, respectively. TieredImageNet contains 779,165 images totally.

CIFAR-FS (Bertinetto et al., 2018) divides the 100 classes in the CIFAR-100 dataset into training, validation, and testing sets, each consisting of 64, 16, and 20 classes, respectively. Each category in CIFAR-FS contains 600 images.

FC100 (Oreshkin et al., 2018a) is also derived from the CIFAR-100 dataset, but the distribution of categories across the training, validation, and testing sets is more diverse, rendering the task more challenging.

## 4.2 IMPLEMENTATION DETAILS

The AdamW optimizer is employed with linear learning rate warm-up along with the cosine scheduler of learning rate. Our experiments are conducted on 8 Nvidia V100 GPUs over a period of 400 epochs. And a multi-crop strategy from DINO is also implemented, which includes 2 global images and 8 local images. Our method is evaluated on three different architectures, namely ResNet18, EfficientNet-b0, and ViT-S (detailed in Appendix.A). For ResNet18, we set the global image resolution to $224 \times 224$ and the local patch resolution to $96 \times 96$. For EfficientNet-b0, we set the global image resolution to $384 \times 384$ and the local resolution to $144 \times 144$. In cascading learning, the training is conducted in two stages. In the first stage, the backbone and the first encoder are trained for 300 epochs. In the second stage, all network components are trained until 400 epochs. The loss weight $\alpha$ in Algorithm 1 is set to 0.1. For the remaining settings, we follow the original self-supervised model implementation.

In our evaluation, a standard evaluation protocol is employed as described in (Mangla et al., 2020; He et al., 2022; Tian et al., 2020b). We construct a cosine classifier to solve few-shot tasks and evaluate our experiments on 5-way 1-shot and 5-way 5-shot classification. For each task, 1 or 5 labeled images are used as support data, and the remaining unlabeled images of the same category are used as query data. In this paper, we sample 2,000 testing tasks for performance evaluation.

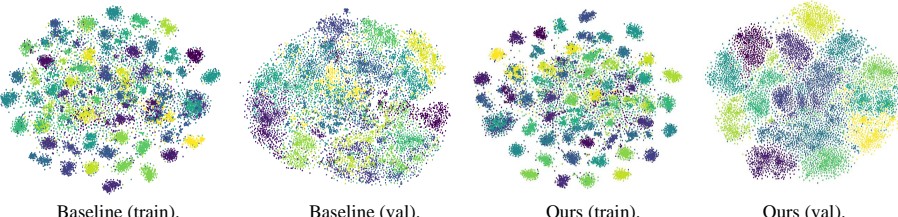

| Baseline (train). | Baseline (val). | Ours (train). | Ours (val). |

Figure 5: Visualization results of features extracted from the network on the miniImageNet. Our scheme has better classification boundaries on the validation set.

## 4.3 COMPARISON

**Results of miniImageNet and tieredImageNet.** Table 1 displays the results obtained on miniImageNet and tieredImageNet. Our proposed method outperforms previous state-of-the-art methods that also use simple CNN (such as ResNet12) on all benchmarks. It is noteworthy that a significant improvement over the second-best method is achieved. Specifically, in the 5-way-1-shot setting, our method outperforms AMTNet (Lai et al., 2022) by 2.07% on miniImageNet, and in the 5-way-5-shot setting, it surpasses MCL (Liu et al., 2022) by 2.91%. The results on MiniImagenet and TieredImageNet are presented in Table 1. Some recent methods use heavyweight backbones like ViT and Swin, which perform well but require a large amount of data to avoid overfitting. While our method with EfficientNet-B0 achieved state-of-the-art performance with better efficiency (in Appendix.A Table.6). Based on our method, modern backbone like EfficientNet have higher performance, but large backbones like ViT only achieve relatively good results (also detailed and analyzed in Appendix.A). Our approach allows the model to better utilize information in the dataset, and improve performance on novel thoughts.

**Results of CIFAR and FC-100.** Table 2 shows the result on CIFAR and FC-100 dataset, we all achieved optimal or suboptimal performance.

Moreover, cross domain results are also carried out and reach SOTA (detailed in Appendix.A).

Overall, the SOTA results are achieved in most of the scenarios in inductive few-shot learning. Although in a few other cases, we only achieved relatively good results. It's still enough to prove the effectiveness and efficiency that ours achieved higher throughput and relatively good performance using a lighter-weight network.

| Method | Backbone | miniImageNet 5-way | | tieredImageNet 5-way | |
|---|---|---|---|---|---|
| | | 1-shot (%) | 5-shot (%) | 1-shot (%) | 5-shot (%) |
| Variational FSL (Zhang et al., 2019) | ResNet-12 | 61.23±0.26 | 77.69±0.17 | - | - |
| MetaOptNet (Lee et al., 2019a) | ResNet-12 | 62.64±0.61 | 78.63±0.64 | 65.99±0.72 | 81.56±0.53 |
| Fine-tuning (Dhillon et al., 2019) | WRN-28-10 | 57.73±0.62 | 78.17±0.49 | 66.58±0.70 | 85.55±0.48 |
| Neg-Cosine (Liu et al., 2020) | ResNet-12 | 63.85±0.81 | 81.57±0.56 | - | - |
| Rethingking-distill (Tian et al., 2020b) | ResNet-12 | 64.82±0.60 | 82.14±0.43 | 71.52±0.69 | 86.03±0.49 |
| Meta-Baseline (Chen et al., 2021) | ResNet-12 | 63.17±0.23 | 79.26±0.17 | 68.62±0.27 | 83.74±0.18 |
| FEAT (Ye et al., 2020) | ResNet-12 | 66.78±0.20 | 82.05±0.14 | 70.80±0.23 | 84.79±0.16 |
| DeepEMD (Zhang et al., 2020) | ResNet-12 | 65.91±0.82 | 82.41±0.56 | 71.16±0.87 | 86.03±0.58 |
| LookingWider (Zhao et al., 2021) | ResNet-12 | 67.96±0.98 | 83.36±0.51 | 73.42±0.95 | 87.72±0.75 |
| ECS (Rizve et al., 2021) | ResNet-12 | 66.82±0.80 | 84.35±0.51 | 71.87±0.89 | 86.82±0.58 |
| PAL (Ma et al., 2021) | ResNet-12 | 69.37±0.64 | 84.40±0.44 | 72.25±0.72 | 86.95±0.47 |
| FRN (Wertheimer et al., 2021) | ResNet-12 | 66.45±0.19 | 82.83±0.13 | 72.06±0.22 | 86.89±0.41 |
| LDA (Xu et al., 2021) | ResNet-12 | 67.76±0.46 | 82.71±0.31 | 71.89±0.52 | 85.96±0.35 |
| SetFeat (Afrasiyabi et al., 2022) | ResNet-12 | 68.32±0.62 | 82.71±0.46 | 73.63±0.88 | 87.59±0.57 |
| MCL (Liu et al., 2022) | ResNet-12 | 69.31±n/a | 85.11±n/a | 73.62±n/a | 86.29±n/a |
| LIF (Li et al., 2021) | ResNet-12 | 68.94±0.28 | 85.07±0.50 | 73.76±0.32 | 87.83±0.59 |
| AMTNet (Lai et al., 2022) | WRN-28 | 70.05±0.46 | 84.55±0.29 | 73.86±0.50 | 87.62±0.33 |
| Baseline | ResNet-12 | 62.74±0.44 | 79.61±0.36 | 70.02±0.43 | 84.66±0.28 |
| Ours | ResNet-12 | **72.12±0.40** | **88.02±0.28** | **77.64±0.48** | **90.41±0.32** |
| FewTURE (Hiller et al., 2022) | Swin-Tiny | 72.40±0.78 | 86.38±0.49 | 76.32±0.87 | 89.96±0.55 |
| HCTransformers (He et al., 2022) | ViT-S×2 | 74.62±0.20 | 89.19±0.13 | 79.57±0.20 | 91.72±0.11 |
| Baseline | EfficientNet-B0 | 62.74±0.44 | 79.61±0.36 | 70.02±0.43 | 84.66±0.28 |
| Ours | EfficientNet-B0 | **74.84±0.36** | **89.84±0.30** | **80.04±0.42** | **92.20±0.31** |

Table 1: Comparison on miniImagenet and tieredImageNet. Bold numbers indicate the best performance, blue number indicates sub-optimal performance. For a fairer comparison, we put the method using the traditional CNN (ResNet12) on top and the method using the new architecture at the bottom.

| Method | Backbone | CIFAR-FS 5-way | | FC100 5-way | |
|---|---|---|---|---|---|
| | | 1-shot (%) | 5-shot (%) | 1-shot (%) | 5-shot (%) |
| Shot-Free (Ravichandran et al., 2019) | ResNet-12 | 69.2±n/a | 84.7±n/a | - | - |
| TEWAM (Qiao et al., 2019) | ResNet-12 | 70.4±n/a | 81.3±n/a | - | - |
| Prototypical Networks (Snell et al., 2017b) | ResNet-12 | 72.2±0.7 | 83.5±0.5 | 37.5±0.6 | 52.5±0.6 |
| MetaOptNet (Lee et al., 2019b) | ResNet-12 | 72.6±0.7 | 84.3±0.5 | 41.1±0.6 | 55.5±0.6 |
| Rethinking (Tian et al., 2020b) | ResNet-12 | 73.9±0.8 | 86.9±0.5 | 44.6±0.7 | 60.9±0.6 |
| ECS (Rizve et al., 2021) | ResNet-12 | 76.8±0.8 | 89.2±0.6 | 47.3±0.8 | 64.4±0.8 |
| PAL (Ma et al., 2021) | ResNet-12 | 77.1±0.7 | 88.0±0.5 | 47.2±0.6 | 64.0±0.6 |
| DeepEMD (Zhang et al., 2020) | ResNet-12 | 74.5±0.3 | 86.4± 0.4 | 45.4± 0.3 | 61.5±0.7 |
| Baseline | ResNet-12 | 70.2±0.4 | 83.0±0.3 | 40.1±0.4 | 58.2±0.4 |
| Ours | ResNet-12 | **78.1±0.4** | **89.9±0.4** | **47.8±0.4** | **65.6±0.4** |
| FewTURE 3 (Hiller et al., 2022) | Swin-Tiny | 77.8±0.8 | 88.9± 0.6 | 47.7± 0.8 | 63.8±0.8 |
| HCTransformers 3 (He et al., 2022) | ViT-S | 78.9±0.2 | 90.5± 0.1 | 48.2± 0.2 | **66.4±0.1** |
| Baseline | EfficientNet-B0 | 71.3±0.4 | 83.2±0.3 | 41.2±0.4 | 58.6±0.4 |
| Ours | EfficientNet-B0 | **79.2±0.4** | **92.0±0.4** | 48.1±0.4 | 66.2±0.4 |

Table 2: Comparison on CIFAR-FS and FC100.

## 4.4 ABLATION STUDY

Table 3 shows the ablation experiments on miniImagenet with ResNet12 as the backbone. MP and DPL bring a great performance improvement by 8.85% on the miniImageNet 5-way-1-shot setting compared with the DINOv2 baseline. GMIX can further improve the accuracy by 1.47% (1-shot). With all these proposed strategies, we surpass the supervised baseline by 9.38% (1-shot). Fig.5 visualizes the embedding space of the dataset. After training with our strategies, the embeddings of

| SL | SSL | DPL | GMIX | Cascading | Selection | 1-shot (%) | 5-shot (%) |
|---|---|---|---|---|---|---|---|
| ✓ | - | - | - | | - | 62.74±0.44 | 79.61±0.36 |
| - | ✓ | - | - | | - | 61.50±0.44 | 78.13±0.36 |
| - | ✓ | ✓ | - | | - | 70.35±0.40 | 85.61±0.39 |
| - | ✓ | ✓ | ✓ | | - | 71.82±0.40 | 86.57±0.37 |
| - | ✓ | ✓ | ✓ | | ✓ | 72.12±0.40 | 88.02±0.28 |

Table 3: Ablation study. Baseline is trained with SL(Supervised learning), we surpass it by a large margin of 9.38% (1-shot) and 8.41% (5-shot).

| Stage of Cascade | DINO | iBot | DINOv2 |
|---|---|---|---|
| $n = 1$ stage | 70.51±0.38 | 71.14±0.37 | 71.82±0.40 |
| $n = 2$ stages | 71.22±0.41 | 71.87±0.37 | **72.12±0.40** |
| $n = 3$ stages | 71.16±0.38 | 71.72±0.40 | 71.78±0.39 |
| $n = 4$ stages | 71.02±0.39 | 71.78±0.41 | 71.89±0.40 |

Table 4: Comparison of *Stage of Cascading Token Selection* and *SSL method* 5-way-1-shot results (%) of miniImageNet.

the validation set become more concentrated and exhibit clearer classification boundaries, indicating that our method has better generalization abilities over novel categories.

**Input Resolution.** For datasets with lower resolution, such as CIFAR, the improvement on performance is not that significant. As pointed out by (He et al., 2022), the patch-based method is not effective on datasets with low resolution. To investigate the impact of resolution on performance, experiments were conducted on different resolution inputs, and the results are shown in Fig.6. This figure also demonstrates that only increasing the resolution may not have positive effects for some methods, as also noted by (He et al., 2022). These experiments suggest that the improvement in performance is not solely attributable to the increase in resolution. Instead, it is the robust backbone trained with our proposed stronger regularization strategies that matter the most in reducing the risk of overfitting.

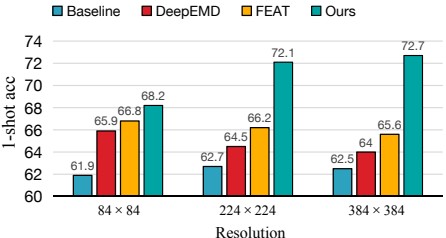

Figure 6: Comparison of different input resolutions.

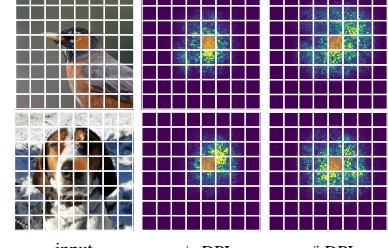

Figure 7: Visualization of local patch token activation regions of two models

| Strategy | Backbone | 1-shot (%) | 5-shot (%) |
|---|---|---|---|
| All | ResNet-12 | 71.44±0.42 | 87.64±0.31 |
| TokenSelection | ResNet-12 | 72.12±0.40 | 88.02±0.28 |

Table 5: Comparison of different token selection strategies on miniImageNet.

**Number of cascading stages and self-supervise methods**: Experiments Table 4 have shown that all self-supervised methods give the best results when number of stage $n$=2. Moreover, among all the self-supervised methods, DINOv2 gives the best results. Therefore, DINOv2 is used in all cases where other results are not specifically mentioned.

**Token Selection Strategy.** Table 5 presents our experiments on validating the design of the token selection strategy. We denote using tokens with higher attention scores for Direct Patch Learning loss as *TokenSelection* and using all the tokens as *All*. In the results of the miniImageNet dataset, it is observed that training with a token selection strategy demonstrates better performance. We attribute such a phenomenon to the irrelevant semantic information contained in other tokens from background regions.

**Activation Region of Direct Patch Learning.** It is observed that the model trained using DPL exhibits a larger activation region for local patch tokens. The receptive field of a deep neural network is theoretically large enough to cover the entire input space. Therefore, using image-level features, such as global pooled features in CNN or the $[cls]$ token in ViT, is adequate to activate the object's entire region, and local tokens are only required to activate their respective local areas. Moreover, directly using global features for training may not benefit generalization to new classes for the model may mainly focus the most significant features on the base classes. When facing novel classes, these features may not be discriminative enough. Through Direct Patch Learning, the model learns different features from one sample more independently. This approach helps avoid overfitting to base classes by learning a more varied set of features.

## 5 CONCLUSIONS

This paper introduces a cascading learning framework for few-shot learning that divides the learning process patch-wise using a token selection. The goal is to enhance the representational capacity of tokens and better leverage the knowledge available from limited data. Our approach demonstrates strong performance with a minimal increase in computational cost. Furthermore, the GMIX modules and patch-wise strategy can be easily integrated into other tasks as plug-and-play modules. Overall, both the insight of the non-annotated part and cascading patch-wise learning strategy is still enlightening to the community.

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

## A  APPENDIX

**The difference between HCT and our method.** 1) HCT He et al. (2022) still did not take the non-annotated part into consideration, while our cascading strategy and self-supervised method are designed to squeeze the residual value of the non-annotated part. 2) As for the annotated part, HCT modifies the DINO to a supervised framework and implements the clustering of patches. Different from clustering, balanced patch weights from our DPL do not emphasize semantic information. 3) Our method avoids the direct cascade of the backbone and achieves relatively good results with fewer parameters in Table.6.

| Method | Params | Throughput | FLOPS | 5-way-1-shot (%) | 5-way-5-shot (%) |
|---|---|---|---|---|---|
| HCT(He et al., 2022) | 42.2M | 170.1 images/s | 20.8G | 74.62±0.20 | 89.19±0.13 |
| Ours | 7.1M | 344.2 images/s | 8.4G | 74.84±0.36 | 89.84±0.30 |

Table 6: Comparison of efficiency. Throughput is measured using the same setting and a V100 GPU.

**Cross domain results.** CUB200 (Wah et al., 2011) is a fine-grained dataset that contains 11,788 images of 200 subcategories belonging to birds. Table 7 presents the result on CUB100. Our strategy gets an improvement of 4.47% on the 1-shot setting and 1.72% on the 5-shot setting compared to the previous SOTA. Table 8 shows the cross-domain performance of miniImageNet to CUB200. We beat baseline by 8.46% on 5-shot and achieved state-of-the-art compared to the previous approach.

| Method | Backbone | 5-way 1-shot | 5-way 5-shot |
|---|---|---|---|
| ProtoNet (Snell et al., 2017a) | ResNet-12 | 66.09±0.92 | 82.50±0.58 |
| RelationNet (Sung et al., 2018) | ResNet-34[†] | 66.20±0.99 | 82.30±0.58 |
| MAML (Finn et al., 2017) | ResNet-34 | 67.28±1.08 | 83.47±0.59 |
| cosine classifier (Chen et al., 2019) | ResNet12 | 67.30±0.86 | 84.75±0.60 |
| MatchNet (Vinyals et al., 2016a) | ResNet-12 | 71.87±0.85 | 85.08±0.57 |
| NegMargin (Liu et al., 2020) | ResNet-18 | 72.66±0.85 | 89.40±0.43 |
| S2M2 (Mangla et al., 2020) | ResNet-34 | 72.92±0.83 | 86.55±0.51 |
| FEAT (Ye et al., 2020) | ResNet-12 | 73.27±0.22 | 85.77±0.14 |
| DeepEMD (Zhang et al., 2020) | ResNet-12 | 75.65±0.83 | 88.69±0.50 |
| Baseline | ResNet-12 | 74.11±0.42 | 85.06±0.29 |
| Ours | ResNet-12 | **80.12±0.40** | **91.12±0.28** |

Table 7: Results on CUB200.

| Method | Backbone | 5-way-5-shot |
|---|---|---|
| ProtoNet (Snell et al., 2017a) | ResNet-12 | 62.02±0.70 |
| MatchingNet (Vinyals et al., 2016a) | ResNet-12 | 53.07±0.74 |
| RelationNet (Sung et al., 2018) | ResNet-12 | 57.71±0.73 |
| Baseline++ (Chen et al., 2019) | ResNet-12 | 65.57±0.70 |
| NegMargin (Liu et al., 2020) | ResNet-12 | 69.30±0.73 |
| Baseline | ResNet-12 | 63.62±0.36 |
| Ours | ResNet-12 | **72.08±0.28** |

Table 8: Cross domain results with miniImageNet being the source dataset and CUB200 being the target dataset.

**ViT comparison on miniImagenet and tieredImageNet.** It indeed failed to achieve SOTA results on ViT, but we have the following analysis of ViT results. 1) In terms of our network, the overall implementation is a cascade strategy, which does not require extremely strong encoding capabilities at the first level of the encoder. However, the result is still valid to prove the effectiveness and efficiency of the strategy by using weak encoders to achieve the SOTA results. 2) The overall analysis shows that the SOTA HCT method He et al. (2022) essentially uses two ViTs, but our ViT method achieves relatively good results with half of the flops and parameters as the HCT.

| Method | Backbone | miniImageNet 5-way | | tieredImageNet 5-way | |
|---|---|---|---|---|---|
| | | 1-shot | 5-shot | 1-shot | 5-shot |
| FewTURE (Hiller et al., 2022) | Swin-Tiny | 72.40±0.78 | 86.38±0.49 | 76.32±0.87 | 89.96±0.55 |
| HCTransformers (He et al., 2022) | ViT-S | 74.62±0.20 | 89.19±0.13 | 79.57±0.20 | 91.72±0.11 |
| Baseline | ViT-S | 56.33±0.39 | 65.61±0.34 | 67.34±0.47 | 79.64±0.29 |
| Ours | ViT-S | 73.36±0.38 | 87.74±0.29 | 78.88±0.45 | 89.07±0.34 |

Table 9: ViT comparison on miniImagenet and tieredImageNet as Fig.1

**EffientNet comparison on miniImagenet.** We managed to reach the SOTA of the miniImagenet and tieredImageNet on EffientNet. And to achieve a fairer comparison, we replaced the backbone of the dominant network architecture with EffientNet for the comparison 10.

| Method | Backbone | miniImageNet 5-way | |
|---|---|---|---|
| | | 1-shot (%) | 5-shot (%) |
| LIF (Li et al., 2021) | EfficientNet-B0 | 67.12±0.36 | 84.12±0.41 |
| AMTNet (Lai et al., 2022) | EfficientNet-B0 | 68.53±0.38 | 84.35±0.29 |
| HCTransformers (He et al., 2022) | EfficientNet-B0 + ViT-S | 69.12±0.21 | 85.51±0.28 |
| Baseline | EfficientNet-B0 | 62.74±0.44 | 79.61±0.36 |
| Ours | EfficientNet-B0 | **74.84±0.36** | **89.84±0.30** |

Table 10: EffientNet comparison on miniImagenet. We adopt their original code with the exception of altering the backbone. For HCTransformer, we follows their setting except modifying the first backbone and replacing the $[cls]$ token with the pooled feature.

**Self-supervised learning.** A multi-stage structure is built on top of the self-supervised learning method. Then here come three self-supervised learning methods, DINO (Caron et al., 2021), DINOv2 (Oquab et al., 2023), and iBot (Zhou et al., 2022).

These methods share a common goal of acquiring a consistent representation, where an identical image, subjected to various data augmentations, is expected to manifest consistent representations, with a predominant emphasis on the foreground patches which have high attention scores. In contrast, our approach differs by introducing a cascading structure. At each level, it progressively eliminates tokens with high attention and instead concentrates on establishing consistency among the remaining tokens. Our primary goal with this approach is to reduce the risk of overfitting to base classes by extracting knowledge from the retained patches. In essence, our aim is to attain representation consistency across multiple contexts.

In addition, those methods are all built upon the foundation of the vision transformer. In contrast, our approach diverges by employing a convolutional neural network as the backbone, augmenting it with 1-4 transformer blocks to serve as the encoder for each cascade. Formally, define $g = f \circ e$ where $f$ is a backbone and $e$ is a transformer encoder. Then $f(x)$ 's outputs are a tensor with shape$[b, c, w, h]$, which is squeezed and then concatenated with a $[cls]$ token to be a tensor with shape $[b, c, h \times w + 1]$, which consisting of $w \times h$ patch tokens and one $[cls]$ tokens. Finally, the encoder takes the concerted tensor as the input.

