# OpenReview forum: "Balanced learning with Token Selection for Few-shot Classification"
_ICLR.cc/2024/Conference — ICLR 2024 Conference Withdrawn Submission_

### Official Review · Reviewer_E3Wj · 2023-10-29

**Soundness:** 3 good
**Presentation:** 2 fair
**Contribution:** 3 good
**Rating:** 5
**Confidence:** 4

**Summary:**

This paper pays attention to the patch-based few-shot learning methods. Motivated by the scenario of not fully leveraging the non-annotated patches in existing methods, this paper proposes to learn all the patches in an image and designs a cascaded multi-stage learning framework, including a direct patch learning strategy, a Gaussian mixup strategy as well as the self-supervised learning strategy. Experiments on multiple benchmark datasets show the effectiveness of the proposed method to some extent.

**Strengths:**

1.	This manuscript is well organized and easy to follow.
2.	The motivation is reasonable and experiments are abundant.

**Weaknesses:**

Several main concerns are as follows:
1. The details of the proposed framework and strategies are not clear, which makes it difficult to convince the readers. Some concerns are as below,
   - The whole objective function to be optimized of the proposed framework is not clear.
   - The details (e.g., architecture and total number of parameters) of the first encoder and following encoders are not clear.
   - How to use and optimize Eq. (1) is not clear.
   - The detail of the self-supervised learning loss used is not clear.
   - The definitions of some notations are not clear. For example, what’s the meaning of z in the Algorithm 1. Is it the same notation used in Figure 4?
   - How to optimize the loss function used in the patch-level label?

2.	From the description in Table 3, this paper focuses on using self-supervised learning to address the few-shot learning problem without using any labels. However, in Figure 4, it seems that the class label is used to obtain the patch-level label. This part should be clear. In addition, the definition of Baseline in the experiments is not clear. Is it just the DINOv2?
3.	The training details of this work is not clear. For example, is the proposed model trained from scratch or trained based on a pre-trained backbone with DINOv2?
4.	The comparisons in Tables 1 and 2 are not strict and fair. For example, many existing FSL methods only use a resolution of 84* 84 for the training images of miniImageNet, while this work uses a much larger resolution. The authors may say they have conducted an experiment in Figure 6. However, how many epochs are used and if the multi-crop strategy is used for other competitors are not clear.
5.	The review of existing works in the literature is not sufficient. For example, the representative patch-based FSL methods, such as [1-4], are not reviewed. Also, for the cascading learning, some existing FSL methods, such as [5-6], have tried the similar ideas.

- [1] Revisiting Local Descriptor based Image-to-Class Measure for Few-shot Learning. CVPR 2019.
- [2] Dense Classification and Implanting for Few-Shot Learning. CVPR 2019.
- [3] Cross attention network for few-shot classification. NeurIPS 2019.
- [4] Joint Distribution Matters: Deep Brownian Distance Covariance for Few-Shot Classification. CVPR 2022.
- [5] Rethinking Few-Shot Image Classification: a Good Embedding Is All You Need? ECCV 2020.
- [6] Self-supervised Knowledge Distillation for Few-shot Learning. BMVC 2021.

**Questions:**

Please refer to the above comments.

---

### Official Review · Reviewer_EnYW · 2023-10-30

**Soundness:** 2 fair
**Presentation:** 3 good
**Contribution:** 3 good
**Rating:** 5
**Confidence:** 4

**Summary:**

The authors present a new approach for few-shot classification based on image patches, which is based upon two main components: 1) Balanced learning of foreground features to obtain a rich representation of the ‘main’ object corresponding to the label, and 2) leveraging the remaining background information to further extract potentially helpful information. To this end, the authors introduce a method to automatically split the tokenised image into ‘foreground’ and ‘background’, followed by individual processing of both sets.
Potential shortcuts in the classification process of foreground objects are reduced via a balanced ‘direct patch learning’ method combined with a Gaussian MixUp Augmentation method to achieve 1), while a cascaded pipeline leveraging self-supervised learning is introduced to tackle 2).
The authors support their method by providing convincing results on the four major few-shot benchmarks (as well as some cross-domain results in the appendix).

**Strengths:**

### Originality & Significance:
-  The idea of selecting a ‘suitable’ subset representing the labelled entity and enforcing a balanced way of learning on this reduced set is original and seems intuitive; Additionally trying to leverage (potentially helpful) information from the ‘background’ area has been shown to help, but is here realized in an interesting iterative way;

### Quality:
-  Approach well motivated, experiments demonstrate convincing results on a variety of benchmarks with two different backbones;
-  Most individual components are well motivated and underlying intuitions for their use are convincing;
-  The presented results and especially ablation (Table 3) indicate the individual contributions of this approach’s components;

### Clarity:
-  Paper is generally easy to read and follow; appropriate use of illustrations
-  Motivation is clearly stated, both for using a balancing strategy for ‘target’ patches to avoid shortcuts, as well as for incorporating the background patches for additional information.

**Weaknesses:**

**Note**: I do think the paper is interesting and ‘almost there’, and I’m happy to update my score if the authors can clarify my questions and provide some insights on these aspects of their work!

---
### Assumptions and Limitations should be clearly stated:
-	The approach heavily relies on the fact that the ‘top tokens’ will actually correspond to the object of interest (‘foreground’), which does not automatically have to be the case -> Imagine a picture of a human with a dog in the park: Both human and/or dog could be ‘foreground’, and might therefore not necessarily correspond to the label;
-	Additionally, what exactly is considered ‘foreground’ and ‘background’ can highly depend on the context in Few-shot settings (i.e. other samples in the support set of the same class) -> See e.g. Hou et al., 2019, _Cross Attention Network for Few-Shot Classification_

&#8594; I'd suggest to clearly state these underlying assumptions and potential limitations in their revised draft;

---
### Ablation & Justification of Token Selection:
Token selection is performed via a concept inspired by PageRank (Sec 3.2), while using the attention matrix as probability transition matrix;

-  Given the provided motivation that DINO’s attention maps focus on the ‘main foreground objects’, this seems quite elaborate/complicated; Is this necessary to obtain a reasonable ranking? What would happen if simply the top-x tokens would be used? And how does the chosen ‘threshold’ of 50% affect results (e.g. vs. 75% or 30%)?
-  The stated motivation holds for ViTs trained using DINO – however, the authors here use a CNN backbone to extract features; This is worth pointing out, and some visualisation (e.g. appendix) of how an attention matrix looks like would be interesting (i.e. does it still hold for representations from CNN-backbones and only few transformer layers, and does it differ between DINO, iBOT and DINOv2 and/or backbones?)
-  Please see the ‘Questions’ part for further questions

---
### Ablation on GMIX:
- Since the authors introduce this ‘new’ mix-up strategy which should generally be applicable as long as two images are available, the question arises how it compares to existing MixUp methods. Ablation is in my opinion required here to provide insight why this method should be used – and would additionally provide insights regarding its value and potentially use beyond the few-shot use case, which could significantly strengthen this contribution.

---
### Minor point: Context of related work

Statements are slightly misleading in some places regarding related work, and might benefit from rewording/clarification: E.g. it is neglected that other works like ‘Doersch et al. (CrossTransformers)’ or ‘Hiller et al.’ are similarly motivated to learn richer representations by using background information (although with the slightly different motivation of using self-supervision in pre-training and/or mixing some episodes to prevent supervision collapse of the representation space);

&#8594;   Note that the paper's introduction makes it sound like the use of background information hasn’t really successfully been explored in the context of (patch-based) few-shot methods – while the related work section doesn’t further clarify this either (and simply mentions the fact that self-supervised has been observed to outperform supervised).

&#8594;  Also the contributions state “prove the potential value of non-annotated parts” – which is true, but still not entirely novel insight.

**Questions:**

Please also see the questions/concerns raised in the 'Weaknesses' part;

---
Token selection via a concept inspired by PageRank (Sec 3.2):
-  I’d like the authors to clarify what exactly is used when setting “s=3”: Is Equation 1 simply used and the Attention matrix cubed, as 's' is shown in the exponent? Or is the state refined using different attention matrices across several layers? &#8594;  Some clarification would be helpful here.

Use of self-supervision:
-  The authors state that “a self-supervised method and loss” is used for the cls token “like DINO, iBOT and DINOv2”; Note that a main difference in iBOT is that a loss is applied to the actual token sequence, not only the ‘cls-token’ – is this done here as well, or not?

Classification at inference time:
-  The authors use “only the output of the first encoder”, and concatenate the “average of foreground tokens and the cls-token”: Is this ‘essential’, or does this simply further boost the performance? And how would only one of them perform (cls vs. avg across patches)?
-  Also: What's the intuition behind this? Usually, we use either an average over the patch tokens (commonly when no cls-token is available) OR the cls-token itself.

Additionally introduced parameters/FLOPs:
-  I’d be curious how many additional parameters & FLOPs are introduced through this method, just to get an impression how it compares to the backbone itself; Some info regarding backbone parameter count vs. the method’s total parameters would be interesting (e.g. in appendix), maybe split into training and inference time.

Minor comments regarding notation & references:
-  For sake of consistency with other (seminal) works and to avoid confusion, it might be easier to stick to the well-established ‘N-way K-shot’ notation (as introduced by Finn et al.) – instead of ‘K-way N-shot’.
-  There exist some duplicates in your references (e.g. Snell et al., Vinyals et al.) that should be corrected.

---

### Official Review · Reviewer_yR4m · 2023-10-30

**Soundness:** 2 fair
**Presentation:** 3 good
**Contribution:** 2 fair
**Rating:** 3
**Confidence:** 5

**Summary:**

This work proposes an end-to-end learning framework for few-shot learning that considers the whole image from a multi-level perspective. The framework includes Direct Patch Learning (DPL) and Gaussian Mixup (GMIX) for balanced learning of annotated subjects, as well as a cascading token selection strategy and self-supervised learning for utilizing knowledge from the non-annotated parts of the image. As the paper illustrates, the proposed method outperforms previous methods in inductive few-shot learning and demonstrates the value of non-annotated parts.
All its motivations are strongly based on the assumption that any patch of the image is beneficial to learning, and it divides the tokens from patches into two sets named top tokens and bottom tokens via a token selection method. And then supervised and self-supervised learning are separately applied to two kinds of tokens.

**Strengths:**

The paper is well organized, tables and figures are easy to read and the results are clearly shown to some extent.

**Weaknesses:**

- The motivation of this work contradicts the findings of a previous paper [1] which shows that image background is harmful to few-shot learning with evidence found by well-designed experiments. Unless the authors prove that the results in [1] are wrong, the motivation of this paper is questionable.
- The techniques taken in this paper are irrelevant to the **few-shot** nature of few-shot learning problems. That is to say, I see nothing not suitable for the proposed methods to be applied to general visual representation learning and classification problems. Thus we cannot learn anything from this paper that can improve our understanding of few-shot learning problems.
- Self-supervised pretraining has proven to be extremely helpful for few-shot learning [2, 3]. As most of the other methods compared in this paper do not use self-supervised pretraining, the comparisons are not fair.
- The resolution of ResNet-12 (or 18?) used in the paper is much higher than previous paper, making comparisons unfair.
- The DINO baseline in this paper is significantly worse than reported in previous paper [3], which shows that DINO has a very high performance on few-shot learning, even approaching the visual encoder of CLIP. This raises the suspicion that all experimental results in this paper may be unreliable.

[1] Rectifying the shortcut learning of background for few-shot learning. NeurIPS 2021.

[2] Self-Supervision Can Be a Good Few-Shot Learner. ECCV 2022.

[3] A Closer Look at Few-shot Classification Again. ICML 2023.

**Questions:**

- What backbone of the ResNet family are you using? ResNet-12 or ResNet-18? They are said in two different places.
- Why use the Markov chain when doing token selection? Actually, the average attention score of cls token is not a strict probability distribution, so why do you insist on using the Markov chain for $\pi_s$?
- In section 3.1, $D_{val}$ data split seems to be ignored when you design the framework and the authors do not state how to determine the hyperparameters.
- Equation (7) is confusing. Do you mean $L^{n^{th}} = L^{{n-1}^{th}}+L_{DPL}+\alpha \times L_{ssl}^{ n^{th}}$?
- In section 4.1, tieredImageNet has 608 classes, not 600. In Figure 4, $w_2$, not $w_x$.
- I cannot evaluate the DPL’s effectiveness owing to an unclear DINO V2 SSL baseline. Why do you choose a dino v2 as the baseline in Table 3 to verify your component effectiveness? Dino-v2 is not comparable because of its dissimilarity with your model.
- Figure 6 shows DeepEmd and feat have lower performance when image resolution is increasing but it is puzzling. Is it because the backbones used for them are still ResNet, not ViT? Or is it because they do not use self-supervised pretraining? What will the results be like if they are equipped with these components?

---

### Official Review · Reviewer_HMyX · 2023-10-30

**Soundness:** 3 good
**Presentation:** 2 fair
**Contribution:** 2 fair
**Rating:** 5
**Confidence:** 3

**Summary:**

In this paper, the author introduces a few-shot classification framework designed to address the limitations of the current patch-wise learning approach. The author presents the "Gaussian-mixup" and "direct patch learning" modules to balance diverse foreground features. Concurrently, a token selection strategy is introduced to extract more information from previously overlooked background details. Ultimately, the proposed method demonstrates a significant improvement over baseline approaches.

**Strengths:**

- For Table 3, the improvement of DPL module over baseline is significant;
- The idea of the paper is simple and effective, and overall speaking, it's technically sound.

**Weaknesses:**

- The sections covering literature review and introduction seem to lack depth in terms of providing sufficient background on patch learning methods. A more comprehensive overview of these methods would aid the reader's understanding and set the context for your subsequent discussions. The paper mentions the process of learning a model using image-level labels for few-shot classification, especially as highlighted in Sec 3.1 "Definition". There appears to be a gap in explaining the transition from image-level to object-level  information utilization for classification. While one can deduce that certain methods might automatically discern important patch tokens from the image and utilize these tokens as feature representation for the ultimate classification loss, this inference should not be left to the reader's assumption. It would enhance the clarity of the paper to provide this background explicitly.
- The subsections on Self-Supervised Learning (SSL) for FSL and cascading learning appear to be cursory. To aid the reader's comprehension and to offer a robust foundation, it's crucial to delve deeper into these areas.
- For Figure 2, the distinction between dashed and solid lines in Figure 2 is unclear. The notations f^{(t)} and f^{(s)} are introduced in the figure but lack an accompanying explanation. The current caption for Figure 2 seems to fall short of giving readers a full understanding of the depicted pipeline. In summary, while Figure 2 appears to be central to the paper's methodology, its current presentation lacks clarity in some aspects.
- Your token selection strategy brings forth the concern of potential noise or irrelevant information being chosen. In methods aiming to identify significant features or tokens, there's always an inherent risk of inadvertently selecting non-representative or misleading tokens.  E.g.,in  Figure 1 Row 2, Using the given example where the class label is "chimpanzee", the inclusion of the human boy in the selected tokens could indeed pose a problem. This selection can mislead the classifier, making it associate features of the human boy with the chimpanzee class. It's essential to clarify how your proposed method deals with such scenarios. Does the method have a mechanism to filter out or down-weight such irrelevant information?

Overall speaking, I have several confusion and concerns of the paper in current stage.

**Questions:**

See weakness part

**Details Of Ethics Concerns:**

N.A.